# PDF: Point Diffusion Implicit Function
# for Large-scale Scene Neural Representation

**Yuhan Ding**[1]*
dingyh22@m.fudan.edu.cn

**Fukun Yin**[1]*
fkyin21@m.fudan.edu.cn

**Jiayuan Fan**[2]†
yfan@fudan.edu.cn

**Hui Li**[2]
lihui21@m.fudan.edu.cn

**Xin Chen**[3]
chenxin2@shanghaitech.edu.cn

**Wen Liu**[3]
liuwen@shanghaitech.edu.cn

**Chongshan Lu**[1]
cslu17@fudan.edu.cn

**Gang Yu**[3]
iskicy@gmail.com

**Tao Chen**[1]
eetchen@fudan.edu.cn

[1] **School of Information Science and Technology, Fudan University, China**
[2] **Academy for Engineering and Technology, Fudan University, China**
[3] **Tencent PCG, Shanghai, China**

## Abstract

Recent advances in implicit neural representations have achieved impressive results by sampling and fusing individual points along sampling rays in the sampling space. However, accurately representing and synthesizing fine-grained textures in unbounded, large-scale outdoor scenes presents a significant challenge, attributable to the exponentially expanding sampling space. To alleviate the dilemma of using individual sampling points to perceive vast expanses, we explore learning the surface distribution of the scene to provide structural priors, thereby reducing the samplable space, and propose a **P**oint **D**iffusion implicit **F**unction, **PDF**, for large-scale scene neural representation. The core of our method is a large-scale point cloud super-resolution diffusion module that enhances the sparse point cloud reconstructed from several training images into a dense point cloud as the explicit prior. Then in the rendering stage, only sampling points with prior points within the sampling radius are retained. That is, the sampling space is reduced from the unbounded space to the scene surface. Meanwhile, to fill in the background details not captured by point clouds, we employ region sampling based on Mip-NeRF 360 for modeling comprehensive background representations. Extensive experiments have demonstrated the effectiveness of our method for large-scale scene novel view synthesis, which outperforms relevant state-of-the-art baselines.

## 1 Introduction

Implicit neural representations have demonstrated proficiency in handling single objects or small scenes and have found extensive applications in the fields of virtual reality [5], 3D reconstruction [21, 31, 36], video generation [7] and computer animation [19, 1, 41] on tasks such as scene representation and new perspective synthesis. Nevertheless, as the scale of the target scene increases, particularly in urban-scale outdoor scenes, traditional implicit neural representation methods encounter significant

---

*Joint first authors.
†Corresponding author.

37th Conference on Neural Information Processing Systems (NeurIPS 2023).

performance limitations. This issue primarily arises from the cubic expansion of the sampling space in larger scenes, rendering it challenging for individual sampling points to cover the entire space.

Fortunately, some methods try to solve this problem through two primary strategies, narrowing the sampling space and expanding the sampling area. The first approach represented by Mega-NeRF [28] decomposes the sampling space into multiple subspaces and models each subspace separately to reduce complexity. But with the scene scale growing, such as reaching the city level, the quantity of these subspaces increases cubically, posing scalability concerns. In contrast, the method represented by Mip-NeRF 360 [2] compresses the sampling space or samples an area instead of a single point so that the sampling space can be filled more easily. Nevertheless, this approach may compromise the precision of the representation.

Benefiting from the inspiration of geometric priors aiding vision tasks, which are widely used in the domain of 3D reconstruction and stereo vision [16, 3, 6]. We are curious whether implicit large scene representations could be made easier with explicit representations. Moreover, for outdoor unbounded large scenes, most of the sampling space is filled with air rather than buildings, cars, plants and other objects that we care about. A reasonable solution is to restrict the large-scale sampling space of the implicit neural representation to the object surface, which is provided by the scene geometry prior. That is to say, we compress a 3D sampling space to a 2D surface plane, which will greatly reduce the representation complexity. At the same time, the network will pay more attention to the foreground, which is the same as the human visual perception system. Of course, for the neglected background information that is relatively less important, we can provide a relatively less accurate expression by compressing the scene to sample the area in space.

In this paper, we propose **PDF**, a **P**oint **D**iffusion implicit **F**unction for large-scale scene neural representation, which learns a dense surface distribution via a diffusion-based point prior generative model to reduce the sampling space. To achieve this, we first explore a large-scale outdoor point cloud augmentation method based on the Point-Voxel Diffusion model [43]. Since point clouds of real outdoor scenes often lack dense ground truth, it is difficult to train a completion module through "sparse-dense" point cloud pairs. Therefore, we initially downsample the point cloud twice, and train a point cloud super-resolution network to generate the denser one from its sparser counterpart. This approach effectively generates dense point clouds in the absence of ground truth data. With the help of the surface point cloud, the sampling points will be retained only if there are reconstruction points within a certain radius, so the space will be greatly reduced to the scene surface. However, the reconstructed point cloud can only model the scene surface and cannot deal with the unbounded background of outdoor scenes. Accordingly, following the concept of NeRF++ [41], we model the foreground and background separately, and use Mip-NeRF 360 [2] to extract background features by sampling regions in the scene space.

Extensive experiments show the effectiveness of our point diffusion implicit function for large-scale scene neural representation, which achieves photo-realistic rendering results and outperforms state-of-the-art methods on OMMO [15] and BlendMVS dataset [35]. We summarize the contributions as follows: **1)** Aiming at novel view synthesis for large outdoor scenes, we propose an implicit neural representation framework based on point diffusion models to provide dense surface priors to cope with the exploding sampling space. **2)** A novel point cloud super-resolution diffusion module is proposed to generate dense surface points from sparse point clouds without dense annotations. **3)** Extensive experiments demonstrate that our PDF network outperforms state-of-the-art methods, including robustness to large-scale outdoor scene representation and the capability to synthesize more photo-realistic novel views. Our code and models will be available.

## 2 Related Work and Background

### 2.1 Implicit Neural Representation

In recent years, Implicit Neural Representation (INR) has witnessed significant advancements and provides a versatile framework for representing complex functions and generating high-dimensional data [23, 17, 26, 37]. By implicitly encoding the scene's appearance and geometry, neural radiance fields enable highly realistic rendering and novel view synthesis [25, 20, 8, 18].

Building upon this foundation, subsequent research has focused on addressing the limitations and pushing the boundaries of INR. Efforts have been made to improve the efficiency and scalability

of neural radiance fields. For instance, Hanocka et al. propose DeepSDF[22], which leverages signed distance functions to implicitly represent 3D shapes. This formulation allows for efficient ray-marching and facilitates tasks such as shape manipulation and interpolation. Furthermore, recent advancements in INR have explored differentiable rendering and differentiable volumetric rendering, enabling the incorporation of geometric and physical priors [10, 4, 27] into the representation. These methods leverage the differentiable nature of neural networks to optimize scene parameters, leading to improved realism and control over the generated content [13, 39, 9, 30]. Another significant extension to the field of INR is PixelNeRF [38]. It extends the capabilities of INR to handle images, going beyond the realm of 3D scenes. PixelNeRF introduces a new differentiable sampler to handle image-based representations, enabling efficient and accurate sampling of pixels from the neural radiance field. In addition to PixelNeRF, Semantic Neural Radiance Fields[12] propose a method to learn scene representations that capture geometry, appearance, and semantic information, facilitating interactive virtual scene editing and content creation.

Overall, these advancements have greatly expanded the capabilities of INR. These developments offer promising avenues for realistic image synthesis, shape completion, scene reconstruction, and dynamic content generation. The ongoing research in this field holds great potential for further advancements in computer graphics, computer vision, and virtual reality applications.

## 2.2 Large-scale Scene Representation

Large-scale scene representation is a crucial aspect of INR research, particularly in the context of computer graphics and computer vision. It involves capturing and modeling complex scenes that encompass extensive spatial extents, such as urban environments, landscapes, or virtual worlds.

One notable work in the domain of large-scale scene representation is Neural Scene Flow Fields[14]. This paper introduces a novel approach to model dynamic scenes at a large scale. The authors propose a scene flow field representation that captures both the geometry and motion of objects in the scene. By leveraging a neural network architecture, they achieve accurate and temporally consistent scene synthesis and reconstruction, even in highly complex and dynamic scenes. The Neural 3D Mesh Renderer[11] is another significant contribution in large-scale scene representation. This work addresses the challenge of representing and rendering detailed 3D meshes of large-scale scenes efficiently. The authors propose a neural network-based renderer that predicts view-dependent textures and geometric details of the scene. This approach enables real-time rendering and interaction with large-scale 3D scenes, opening up possibilities for interactive virtual reality experiences and immersive simulations. In addition to these works, Mega-NeRF [28] and Bungee-NeRF [33] are two other notable approaches based on the neural radiance field for constructing interactive 3D environments from large-scale visual captures. They address the challenges of modeling and rendering large-scale scenes, spanning from buildings to multiple city blocks and utilizing thousands of images captured from drones. They extend the capabilities of NeRF to handle multi-scale rendering, capturing various levels of detail and enabling the interactive exploration of diverse 3D environments.

Overall, the field of large-scale scene representation within INR has witnessed significant progress. These contributions have paved the way for realistic, interactive, and semantically meaningful representations of expansive virtual environments, urban landscapes, and dynamic scenes. The ongoing research in this area holds great potential for further advancements in computer graphics, virtual reality, and immersive simulations.

## 3 Methodology

In this paper, we aim to develop a novel point diffusion model implicit function to reduce the sampling space and improve the ability to represent large-scale scenes ($c.f.$ Fig. 1). Our PDF network mainly consists of two modules, a diffusion-based component for point cloud super-resolution and foreground rendering, and a region-sampling module focused on background processing. The former introduces a diffusion model to enhance the sparse point cloud reconstructed from the input images into a dense point cloud, which provides optional points in the rendering stage to reduce the sampling space ($c.f.$ Sec. 3.1). The latter samples regions rather than individual points from unbounded scenes so that it is easy to fill sampled regions and complement the background for new viewpoint synthesis ($c.f.$ Sec. 3.2). In the final subsection, implementation details and losses are elaborated ($c.f.$ Sec. 3.3).

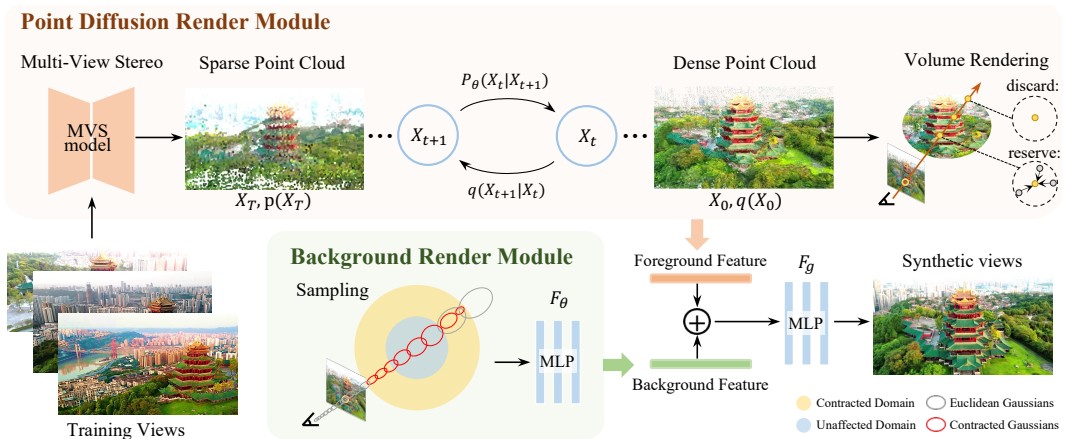

Figure 1: The pipeline of our point diffusion implicit function. Our method consists of two modules, a point diffusion rendering module and a background rendering module. The former learns the surface distribution of the scene through a diffusion-based point cloud super-resolution model and renders foreground features from the dense point cloud surface. The latter follows Mip-NeRF 360's strategy to render background features. Finally, the foreground and background features are fused to generate photo-realistic novel views for large-scale outdoor scenes.

## 3.1 Point Upsampling Diffusion

In this section, we introduce our large-scale outdoor point cloud super-resolution module based on a denoising diffusion probabilistic model ($c.f.$ Fig. 2).

**Point Cloud Pair Preparation.** Due to the lack of dense large-scale outdoor point cloud ground truth, we need to train a diffusion-based super-resolution network to sample a dense surface, symbolized as $x_d \in \mathbb{R}^{N \times 3}$, from the point cloud reconstructed by COLMAP [24], denoted as $x_s \in \mathbb{R}^{M \times 3}$. Concurrently, to mitigate the risk of over-fitting, the point cloud reconstructed from the training views is not utilized as the ground truth for the diffusion model. Instead, training data comprises pairs of the sparse point cloud $z_0 \in \mathbb{R}^{n \times 3}$ and the sparser point cloud $x_0 \in \mathbb{R}^{m \times 3}$, adhering to $m < n < M < N$. More specifically, we downsample the sparse point cloud $x_s$ reconstructed by COLMAP to get an even sparser point cloud $z_0$. Then we further downsample $z_0$ to get the sparsest point cloud $x_0$, where $x_s$, $z_0$ and $x_0$ have progressively sparser relationships. Our training process recovers $z_0$ from the sparsest $x_0$. During testing, we take $x_s$ as input to generate a denser super-resolved point cloud $x_d$.

**Point Super-resolution Diffusion.** Our point super-resolution denoising diffusion probabilistic model is a generative model, which starts with Gaussian noise and progressively denoises to generate scene structure priors. We record the output containing different levels of noise produced by each step as $\hat{x}_T, \hat{x}_{T-1}, ..., \hat{x}_0$, where $\hat{x}_T$ is sampled from Gaussian noise, and $\hat{x}_0$ represents the generated point cloud with dense surface. Since we already have a sparse point cloud prior $z_0$, our target point cloud can be denoted as $x_0 = (z_0, \hat{x}_0)$ and the intermediate point cloud during the denoising process can be denoted as $x_t = (z_0, \hat{x}_t)$. Subsequently, we define a point super-resolution diffusion process involving a prior shape $z_0$, consisting of a forward process and a backward process.

Forward Process. Gaussian noise is repeatedly added to the original point cloud $x_0$, resulting in a series of noisy point clouds $x_1, x_2, ..., x_T$:

$$q(\hat{x}_t | \hat{x}_{t-1}, z_0) \sim \mathcal{N}(\hat{x}_t; \sqrt{1 - \beta_t}\hat{x}_{t-1}, \beta_t I) \tag{1}$$

where $\beta_t$ represents a pre-defined increasing sequence of Gaussian noise values, which dictates the magnitude of noise incrementally introduced at each step of the process.

Reverse Process. Given a point cloud with more noise $x_t$, reverse the forward process and find the posterior distribution for a less noisy one $x_{t-1}$:

$$p_\theta(\hat{x}_{t-1} | \hat{x}_t, z_0) \sim \mathcal{N}(\mu_\theta(x_t, z_0, t), \sigma_t^2 I) \tag{2}$$

where $\mu_\theta(x_t, z_0, t)$ is the predicted shape at $t - 1$ step.

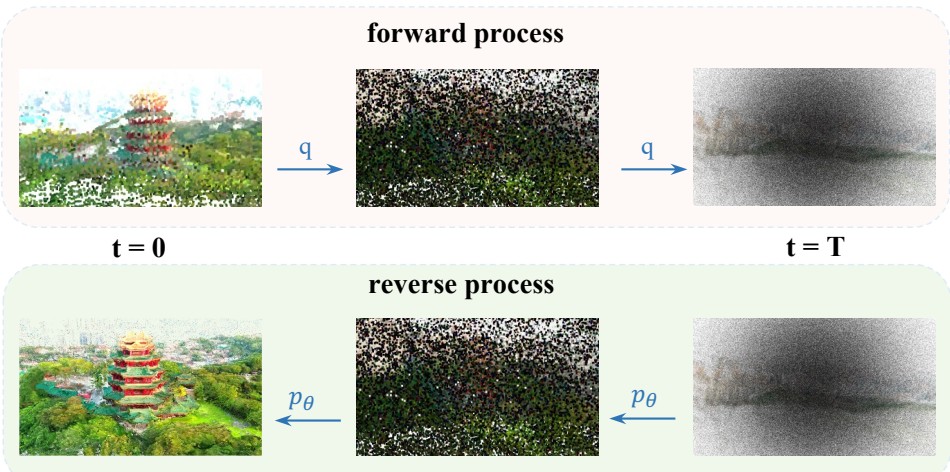

Figure 2: Our point upsampling diffusion. In the forward process, Gaussian noise is gradually added to the sparse point cloud. In the reverse process, the noise is gradually removed to obtain a dense point cloud surface.

Therefore, our point cloud upsampling diffusion model can be regarded as a noise adding and denoising process. The former gradually adds random noise to the initial point cloud $x_0$ through the forward process; the latter denoises sequentially through the reverse process to obtain a dense point cloud $x_0$. Based on Markov transition probabilities, the whole process can be expressed as:

$$q(\hat{x}_{0:T}, z_0) = q(\hat{x}_0, z_0) \prod_{t=1}^{T} q(\hat{x}_t | \hat{x}_{t-1}, z_0) \tag{3}$$

$$p_\theta(\hat{x}_{0:T}, z_0) = p(\hat{x}_T, z_0) \prod_{t=1}^{T} p_\theta(\hat{x}_{t-1} | \hat{x}_t, z_0) \tag{4}$$

Throughout the optimization process, our prior shape $z_0$ is fixed, and only the missing surface point cloud is diffused. The network is typically trained with a simplified $L2$ denoising loss:

$$\mathcal{L}_{\mathcal{D}} = ||\epsilon - \epsilon_\theta(\hat{x}_t, z_0, t)||^2 \tag{5}$$

where $\epsilon$ is the added random noise and $\epsilon \sim \mathcal{N}(0, I)$, and $\epsilon_\theta(\hat{x}_t, z_0, t)$ is the prediction noise output. Since point cloud prior $z_0$ is fixed, it will be masked when minimizing the loss.

### 3.2 Volume Rendering and Implicit Function Representation

**Foreground Rendering.** For the foreground, we sample points along the rays from the dense point cloud $x_0$ and render features in the neighborhood, following Point-NeRF [34]. The difference is that our point super-resolution diffusion module only generates denser point cloud coordinates without color information, so we redesign a more general point aggregate module. For ray marching through a pixel, we sample $M$ sampling points at $\{p_i \mid i = 1, ..., M\}$, and query $K$ neighboring neural points $kp_i = \{kp_i^1, kp_i^2, ..., kp_i^K\}$ around $p_i$ within a certain euclidean distance radius $R$. Then we interpret the local geometric structure as a feature $f_i$ of each sampling point $p_i$ to equip structural information. Therefore, we utilize $c_i$ and $kc_i$ to represent the coordinates of the sampling point and its neighborhood points, respectively. The geometric structure feature $kf_i = \{kf_i^1, kf_i^2, ..., kf_i^K\}$ of the neighborhood are encoded as follows:

$$kf_i = MLP(c_i \oplus kc_i \oplus (c_i - kc_i) \oplus d(c_i, kc_i)) \tag{6}$$

where $d(,)$ is the Euclidean distance between two points, and $\oplus$ is the concatenation operator. Next, the local geometric features $f_i$ of the sampling points $p_i$ are obtained by neighborhood points $kp_i$ weighted summation:

$$f_i = SUM(softmax(MLP(kf_i)) \odot kf_i) \tag{7}$$

where softmax operation is performed on each dimension, and $\odot$ is the hadamard product. Our point-based radiance field can be abstracted as a neural module that regresses the volume density $\sigma$ and view-dependent radiance $r$ from coordinates $c$, local geometric features $f$, and ray direction $d$ according to Point-NeRF [34]:

$$(\sigma, r) = Point\text{-}NeRF(c, d, f) \tag{8}$$

Finally, the foreground feature is synthesized by each neural sampling point along the sampling ray.

**Background Rendering.** In consideration of the limitation that point clouds are confined to representing foreground elements and are incapable of addressing unbounded background contexts, it becomes necessary to procure supplementary background features. Benefiting from Mip-NeRF 360 [2], which contracts the scene to a bounded ball and then samples a region to meet the challenge of large scenes, we employ this method to extract background features as a supplement along the same sampling ray as Point-NeRF [34].

**Fore-Background Fusion.** Since the detail-preserving foreground features can be obtained from the dense surface points, while the bounded domain can cope with large scenes but loses details during the compression process. So we propose a foreground-background fusion module consisting of several layers of multi-layer perceptrons to preserve their respective advantages.

We adopt $L2$ loss to supervise our rendered pixels $r_p$ from ray marching with the ground truth $r_g$, to optimize our PDF volume render reconstruction network.

$$\mathcal{L}_\mathcal{R} = ||r_p - r_g||^2 \tag{9}$$

### 3.3 Implementation Details

Our PDF method is a two-stage neural representation network for outdoor unbounded large-scale scenes. We optimize these two stages separately.

In the first stage, a diffusion-based point cloud super-resolution network is designed to learn a prior distribution to generate a dense point cloud surface. In the point cloud pair preparation process, we employed the random down-sampling method with a retention rate between 0.2 and 1 for both samplings. For point super-resolution diffusion, we set $T = 1000$, $\beta_0 = 10^{-4}$, $\beta_T = 0.01$ and linearly interpolate other $\beta$'s for all experiments. We use Adam optimizer with learning rate $2 \times 10^{-4}$ and train on 4 A100 GPUs for around one day.

In the second stage, the foreground and background extraction modules plus a feature fusion module are optimized. We find 8 neighbors for each sampling point and expand the dimension of neighborhood geometric features to 8. Both the foreground and the background output a 128-dimensional feature, and then they are concatenated and passed through 4 MLP layers to get the color of the rendered point. We train this stage using Adam optimizer with an initial learning rate $5 \times 10^{-4}$ for $2 \times 10^6$ iterations about 20 hours on a single A100 GPU.

## 4 Experiments

### 4.1 Experimental settings

**Dataset.** We use two outdoor large-scale scene datasets, OMMO [15] and BlendedMVS [35], to evaluate our model. The OMMO dataset is a real fly-view large-scale outdoor multi-modal dataset, containing complex objects and scenes with calibrated images, prompt annotations and point clouds. The number of training point cloud samples in the OMMO dataset varies from 40,000 to 100,000 for different scenes, including abundant real-world urban and natural scenes with various scales, camera trajectories, and lighting conditions. More experimental results can be found in our supplementary material.

**Baselines and Evaluation Metrics.** We compare our method with the previous state-of-the art methods on novel view synthesis, including NeRF [19], NeRF++ [41], Mip-NeRF [1], Mip-NeRF 360 [2], Mega-NeRF [28], Ref-NeRF [29]. NeRF is the first continuous MLP-based neural network for synthesizing photo-realistic views of a scene through volume rendering. NeRF++ models large-scale unbounded scenes by separately modeling foreground and background neural representations.

Mip-NeRF reduces aliasing artifacts and better represents fine details by using anti-aliasing cone sampling. Mip-NeRF 360 models large unbounded scenes using non-linear scene parameterization, online distillation, and distortion-based regularization. Mega-NeRF uses a sparse structure and geometric clustering algorithm to decompose the scenes. Ref-NeRF improves synthesized views by restructuring radiance and regularizing normal vectors. To evaluate the performance of each method for large-scale implicit neural representation, we use three common metrics for novel view synthesis: Peak Signal-to-Noise Ratio (PSNR), Structural Similarity (SSIM [32]), and Learned Perceptual Image Patch Similarity (LPIPS [42]). Higher PSNR and SSIM indicate better performance, while lower LPIPS indicates better performance.

## 4.2 Performance Comparison

**Quantitative Results.** Quantitative comparisons on the OMMO [15] dataset are shown in Tab. 1, including the mean PSNR, SSIM, and LPIPS. We outperform other methods on all average evaluation metrics, especially LPIPS, a perceptual metric close to the human visual system, which is significantly more sensitive to the foreground than the background. So better LPIPS metric indicates that our model can better reconstruct the foreground of the scene, benefiting from sampling the foreground from the reconstructed dense point cloud surface instead of the entire sampling space.

NeRF [19], NeRF++ [41], Mip-NeRF [1], and Ref-NeRF [29] are not specially designed for large-scale scenes, so directly applying them to large scenes will lead to performance degradation. Mip-NeRF 360 [2] and Mega-NeRF [28] have achieved the optimal performance in one or several scenes by sampling regions in the limited sampling space or subdividing the sampling space. But it is still not as good as ours in most scenes due to the loss of detail caused by compression or decomposing the sampling space.

Table 1: Quantitative results of our PDF method with the baselines on the OMMO dataset. ↑ means the higher, the better.

| Scene ID | NeRF[19] | | | NeRF++[41] | | | Mip-NeRF[1] | | | Mip-NeRF 360[2] | | | Mega-NeRF[28] | | | Ref-NeRF[29] | | | Ours | | |
|---|---|---|---|---|---|---|---|---|---|---|---|---|---|---|---|---|---|---|---|---|---|
| | PSNR↑ | SSIM↑ | LPIPS↓ | PSNR↑ | SSIM↑ | LPIPS↓ | PSNR↑ | SSIM↑ | LPIPS↓ | PSNR↑ | SSIM↑ | LPIPS↓ | PSNR↑ | SSIM↑ | LPIPS↓ | PSNR↑ | SSIM↑ | LPIPS↓ | PSNR↑ | SSIM↑ | LPIPS↓ |
| 1 | **16.93** | **0.37** | **0.744** | 16.86 | 0.36 | 0.780 | 16.84 | **0.37** | 0.793 | 13.91 | 0.31 | 0.771 | 16.12 | 0.34 | 0.782 | 15.10 | 0.34 | 0.755 | 14.80 | 0.32 | 0.755 |
| 2 | 15.31 | 0.44 | 0.694 | 14.89 | 0.47 | 0.653 | 15.16 | 0.40 | 0.731 | 15.06 | 0.44 | 0.646 | 15.64 | 0.47 | 0.679 | 15.90 | 0.49 | 0.632 | **19.63** | **0.62** | **0.374** |
| 3 | 14.38 | 0.28 | 0.556 | 14.64 | 0.29 | 0.547 | 14.56 | 0.29 | 0.533 | 15.21 | 0.33 | 0.517 | **15.44** | **0.37** | 0.526 | 14.74 | 0.34 | **0.515** | | | |
| 4 | 25.39 | 0.86 | 0.431 | 27.47 | 0.90 | 0.380 | 21.78 | 0.76 | 0.469 | 27.68 | **0.94** | 0.292 | 23.36 | 0.86 | 0.419 | 27.86 | 0.91 | 0.404 | **31.74** | **0.94** | **0.202** |
| 5 | 22.26 | 0.67 | 0.531 | 24.32 | 0.73 | 0.450 | 14.98 | 0.54 | 0.633 | 25.78 | 0.80 | 0.317 | 23.54 | 0.76 | 0.436 | 23.54 | 0.71 | 0.491 | **27.58** | **0.90** | **0.162** |
| 6 | 24.09 | 0.68 | 0.504 | 25.59 | 0.75 | 0.396 | 23.18 | 0.66 | 0.529 | **28.86** | **0.90** | **0.211** | 24.92 | 0.77 | 0.393 | 26.07 | 0.72 | 0.459 | 23.69 | 0.87 | 0.212 |
| 7 | 5.36 | 0.17 | 0.747 | 21.93 | 0.71 | 0.542 | 15.57 | 0.64 | 0.624 | 23.05 | 0.73 | 0.523 | 22.33 | 0.69 | 0.552 | **25.79** | 0.73 | 0.511 | 21.46 | **0.81** | **0.193** |
| 8 | 21.14 | 0.50 | 0.594 | 22.91 | 0.57 | 0.509 | 19.82 | 0.46 | 0.638 | 25.07 | 0.71 | 0.354 | 16.65 | 0.48 | 0.431 | 21.21 | 0.49 | 0.606 | **27.62** | **0.92** | **0.101** |
| 9 | 14.92 | 0.34 | 0.744 | 14.57 | 0.34 | 0.732 | 14.58 | 0.34 | 0.746 | 15.40 | 0.30 | 0.706 | 17.32 | **0.49** | 0.673 | **20.34** | 0.43 | 0.649 | 15.77 | **0.49** | **0.381** |
| 10 | 22.26 | 0.55 | 0.626 | 24.37 | 0.60 | 0.578 | 19.80 | 0.53 | 0.643 | **26.68** | 0.72 | 0.420 | 21.78 | 0.62 | 0.558 | 24.23 | 0.58 | 0.597 | 25.74 | **0.83** | **0.136** |
| 11 | 22.36 | 0.82 | 0.420 | 24.61 | 0.85 | 0.342 | 22.81 | 0.82 | 0.423 | 27.06 | 0.93 | 0.217 | 24.37 | 0.84 | 0.392 | 23.81 | 0.84 | 0.355 | **30.29** | **0.95** | **0.188** |
| 12 | 22.41 | 0.59 | 0.533 | 24.29 | 0.68 | 0.447 | 22.13 | 0.60 | 0.526 | 28.12 | 0.83 | 0.274 | 21.60 | 0.62 | 0.493 | 23.06 | 0.60 | 0.524 | **27.92** | **0.86** | **0.063** |
| 13 | 22.27 | 0.59 | 0.608 | 23.52 | 0.62 | 0.581 | 18.90 | 0.54 | 0.673 | **26.63** | **0.77** | 0.403 | 25.50 | 0.72 | 0.517 | 23.29 | 0.61 | 0.594 | 25.94 | 0.74 | **0.205** |
| 14 | 19.85 | 0.55 | 0.569 | 23.89 | 0.74 | 0.417 | 17.06 | 0.48 | 0.655 | 28.06 | 0.89 | 0.224 | 24.42 | 0.75 | 0.411 | 21.76 | 0.63 | 0.508 | **28.11** | **0.94** | **0.127** |
| 15 | 20.35 | 0.53 | 0.552 | 21.71 | 0.61 | 0.490 | 19.44 | 0.49 | 0.594 | **28.63** | **0.89** | 0.179 | 22.69 | 0.67 | 0.445 | 20.33 | 0.50 | 0.576 | 27.22 | **0.89** | **0.136** |
| 16 | 17.86 | 0.40 | 0.631 | 18.75 | 0.41 | 0.597 | 18.49 | 0.40 | 0.610 | 10.01 | 0.34 | 0.850 | **20.26** | **0.53** | 0.509 | 19.64 | 0.43 | 0.572 | 18.70 | 0.47 | **0.392** |
| 17 | 22.02 | 0.57 | 0.610 | 24.20 | 0.67 | 0.461 | 17.01 | 0.53 | 0.696 | **29.53** | 0.83 | 0.247 | 17.23 | 0.57 | 0.529 | 23.17 | 0.59 | 0.529 | 26.59 | **0.88** | **0.111** |
| 18 | 26.06 | 0.75 | 0.428 | 23.88 | 0.64 | 0.461 | 24.61 | 0.73 | 0.469 | **28.55** | 0.86 | 0.265 | 24.76 | 0.73 | 0.448 | 22.79 | 0.67 | 0.569 | 28.07 | **0.91** | **0.152** |
| 19 | 14.20 | 0.40 | 0.726 | 13.86 | 0.37 | 0.703 | 13.84 | 0.39 | 0.738 | 14.72 | 0.37 | 0.676 | 23.81 | 0.68 | 0.465 | 14.34 | 0.39 | 0.691 | **27.55** | **0.84** | **0.170** |
| 20 | 22.84 | 0.61 | 0.499 | 23.28 | 0.64 | 0.475 | 22.41 | 0.60 | 0.519 | 21.11 | 0.60 | 0.490 | 21.76 | 0.63 | 0.508 | 23.62 | 0.50 | 0.574 | **26.88** | **0.81** | **0.197** |
| 21 | 22.59 | 0.51 | 0.532 | 21.84 | 0.47 | 0.593 | 22.31 | 0.51 | 0.537 | 25.64 | 0.75 | 0.344 | 21.92 | 0.51 | 0.578 | 21.07 | 0.44 | 0.672 | **28.62** | **0.94** | **0.141** |
| 22 | 16.53 | 0.47 | 0.733 | 20.66 | 0.56 | 0.575 | 13.37 | 0.42 | 0.776 | 24.79 | 0.77 | 0.362 | 20.84 | 0.60 | 0.527 | 20.31 | 0.53 | 0.615 | **26.33** | **0.85** | **0.074** |
| 23 | 18.99 | 0.41 | 0.669 | 19.51 | 0.42 | 0.597 | 18.09 | 0.39 | 0.671 | 21.25 | 0.51 | 0.539 | 20.13 | 0.44 | 0.585 | 19.94 | 0.41 | 0.622 | **21.64** | **0.65** | **0.206** |
| 24 | 19.32 | 0.39 | 0.696 | 23.14 | 0.52 | 0.535 | 16.89 | 0.37 | 0.715 | 25.86 | 0.71 | 0.373 | 23.87 | 0.56 | 0.518 | 22.17 | 0.45 | 0.616 | **30.90** | **0.87** | **0.097** |
| 25 | 24.72 | 0.55 | 0.528 | 22.42 | 0.51 | 0.613 | 24.24 | 0.54 | 0.542 | 25.98 | 0.63 | 0.457 | 23.62 | 0.50 | 0.598 | | | | **30.85** | **0.94** | **0.083** |
| 26 | 8.56 | 0.24 | 0.564 | 19.94 | 0.59 | 0.513 | 13.43 | 0.35 | 0.688 | 14.59 | 0.46 | 0.626 | 19.23 | 0.67 | 0.467 | 21.00 | 0.62 | 0.489 | **23.88** | **0.83** | **0.311** |
| 27 | 4.54 | 0.01 | 0.705 | 21.25 | 0.55 | 0.564 | 14.82 | 0.45 | 0.674 | | | 0.235 | 20.82 | 0.52 | 0.590 | | | | **21.77** | **0.66** | **0.164** |
| 28 | 24.48 | 0.66 | 0.479 | 23.28 | 0.64 | 0.475 | 24.76 | 0.66 | 0.406 | **29.62** | 0.87 | 0.240 | 25.87 | 0.72 | 0.442 | 22.17 | 0.45 | 0.616 | 29.22 | **0.91** | **0.153** |
| 29 | 22.98 | 0.61 | 0.540 | 23.17 | 0.62 | 0.529 | 23.01 | 0.61 | 0.539 | 25.51 | 0.74 | 0.400 | 21.57 | 0.61 | 0.557 | 21.11 | 0.54 | 0.631 | **25.86** | **0.84** | **0.174** |
| 30 | 20.23 | 0.52 | 0.605 | 23.27 | 0.64 | 0.476 | 18.63 | 0.46 | 0.675 | **26.54** | 0.84 | 0.296 | 24.04 | 0.69 | 0.459 | 21.62 | 0.54 | 0.586 | 26.10 | **0.93** | **0.096** |
| 31 | 18.97 | 0.37 | 0.645 | 19.05 | 0.37 | 0.643 | 18.91 | 0.36 | 0.659 | 13.08 | 0.23 | 0.708 | 20.93 | 0.60 | 0.545 | 19.18 | 0.37 | 0.645 | **26.68** | **0.90** | **0.208** |
| 32 | 17.99 | 0.58 | 0.621 | 18.99 | 0.61 | 0.540 | 11.28 | 0.42 | 0.687 | 11.29 | 0.57 | 0.601 | 21.29 | **0.70** | 0.475 | 18.98 | 0.60 | 0.565 | **23.43** | 0.69 | **0.142** |
| 33 | 5.79 | 0.01 | 0.745 | 20.19 | 0.50 | 0.597 | 14.31 | 0.42 | 0.755 | 22.76 | 0.63 | 0.457 | 22.89 | 0.64 | 0.478 | 21.23 | 0.52 | 0.578 | **22.91** | **0.75** | **0.134** |
| **Mean** | 18.72 | 0.48 | 0.600 | 21.45 | 0.58 | 0.538 | 18.39 | 0.50 | 0.623 | 23.10 | 0.67 | 0.419 | 21.63 | 0.62 | 0.508 | 21.28 | 0.55 | 0.574 | **25.10** | **0.79** | **0.205** |

**Qualitative Results.** Qualitative results on the OMMO [15] dataset are shown in Fig. 3. We can see that the rendering results of NeRF [19] and Mip-NeRF [1] are of the lowest quality, and they use global MLPs for the entire space to reconstruct radiance fields, resulting in a trade-off in the accuracy of sampling the foreground and background which makes them almost impossible to handle large-scale unbounded scenes. NeRF++ [41], Mega-NeRF [28] and Ref-NeRF [29] improve some limitations of NeRF by corresponding techniques, but the rendering results are often missing details, especially when the scene contains a lot of intricate details. The rendering quality of Mip-NeRF 360 [2] is relatively high, but loses some detail and edges due to its down-scaling of the scene into a limited sampling space. Our method uses the dense point cloud up-sampled by the diffusion model as a detailed foreground geometry prior combined with Mip-NeRF 360 background features, so our model can reconstruct the fine foreground texture provided by the generative model. At the same

**NeRF**

**NeRF++**

**Mip-NeRF**

**Mip-NeRF 360**

**Mega-NeRF**

**Ref-NeRF**

**Ours**

**GT**

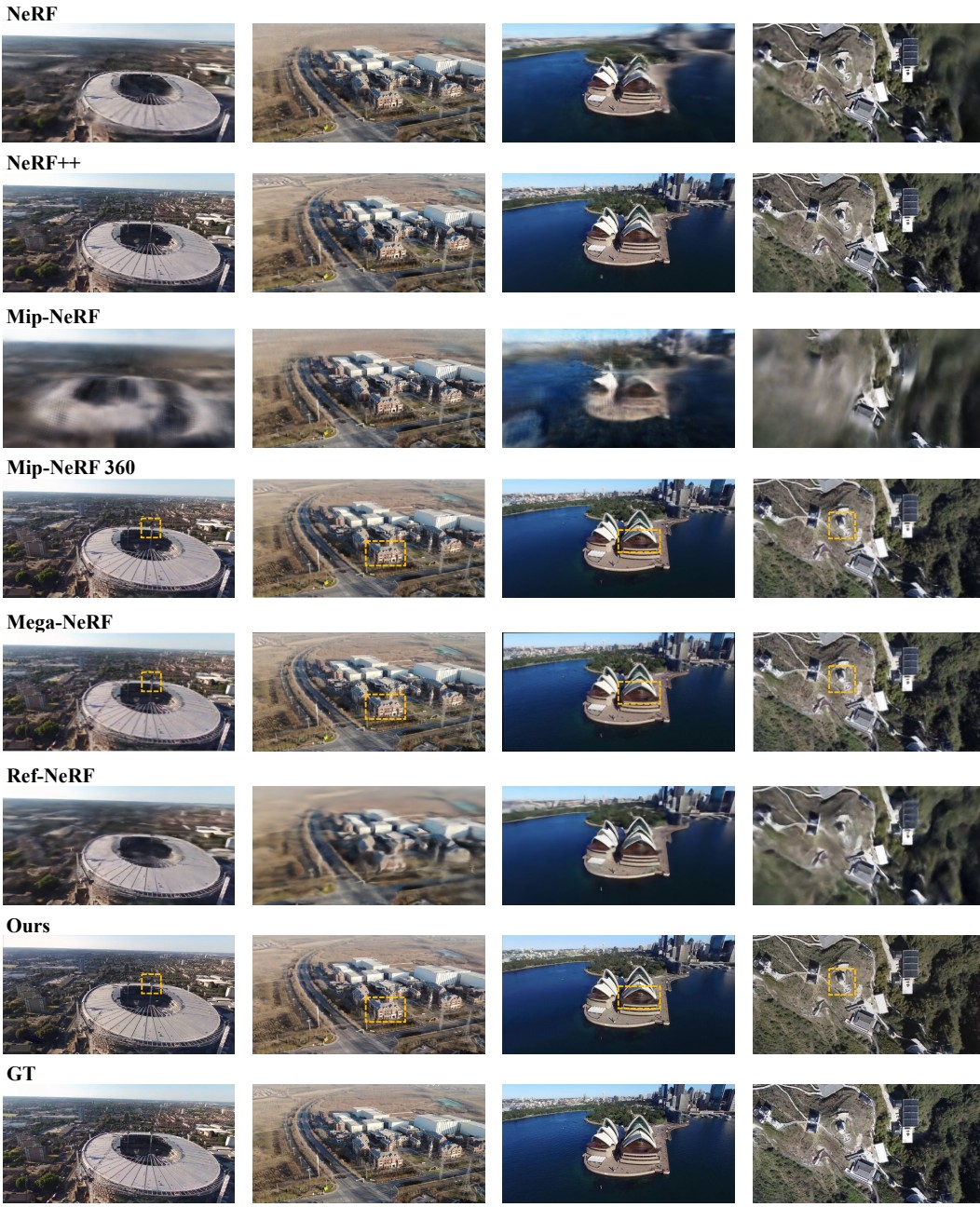

Figure 3: Qualitative results of our method with the baselines on the OMMO dataset. Our PDF method outperforms baseline methods with reliably constructed details. For Mip-NeRF and Mega-NeRF, which are also aimed at large scenes, we use yellow dashed boxes to mark some areas that are easy to distinguish the performance of details. Please zoom-in for the best of views.

time, compared with Mip-NeRF 360, our method is more robust to scene representation and new view generation without failing scenes ($c.f$. Fig. 4).

## 4.3 Ablation Studies

We perform multiple ablation studies to validate the effectiveness of our proposed modules. Tab. 2 shows the impact of diffusion point cloud super-resolution module and background feature fusion module on the 5-th scene (sydney opera house) from the OMMO dataset [15].

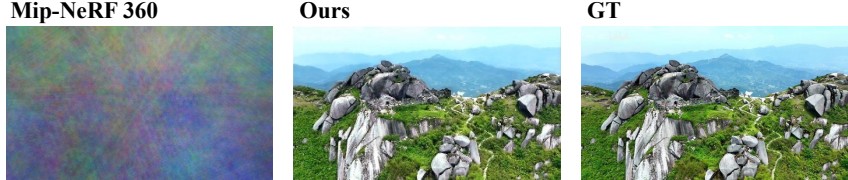

Figure 4: A failure scene representation of Mip-NeRF 360.

Table 2: Quantitative performance of ablation experiments, including removing both the diffusion-based point cloud up-sampling module and the background fusion module, removing only the diffusion-based point cloud up-sampling module, removing only the background fusion module, our PDF method.

| Method | PSNR↑ | SSIM↑ | LPIPS↓ |
|---|---|---|---|
| w/o diffusion, w/o background | 9.28 | 0.51 | 0.355 |
| w/o diffusion, w/ background | 21.05 | 0.83 | 0.219 |
| w/ diffusion, w/o background | 22.93 | 0.78 | 0.235 |
| **Ours** | **27.58** | **0.90** | **0.162** |

For the ablation experiment on the effectiveness of diffusion, we remove the diffusion-based point cloud up-sampling module and sample directly on the sparse point cloud reconstructed by COLMAP [24] from training views. Since the directly reconstructed point cloud is very sparse and concentrated in the central area, only a very blurry image with large missing blocks can be rendered, as shown in the first column of Fig. 5. At the same time, quantitative indicators also suggest that this method is not suitable for outdoor unbounded large-scale scenes with its PSNR of 9.28.

For the ablation experiment on the effectiveness of background fusion, we remove the background fusion module and render novel view images directly from the diffusion-enhanced point cloud. As shown in the second column of Fig. 5, with the help of the dense point cloud produced by the diffusion module learning the scene distribution, we find that large missing patches have been filled in and produce a more refined foreground. However, limited by the characteristics of point cloud expression, the background points are very sparse, which leads to blurred background rendering results. Quantitative results, while substantially improved, still convey poor image quality.

As shown in the third column of Fig. 5, using the background fusion module alone can also fill in the missing blocks of the background, but due to the sparseness of the point cloud reconstructed by COLMAP [24], it will lead to the loss of detail and blurring of the rendering result. However, our method, which combines a diffusion module and a background fusion module, achieves satisfactory quantitative and qualitative performance and surpasses existing methods.

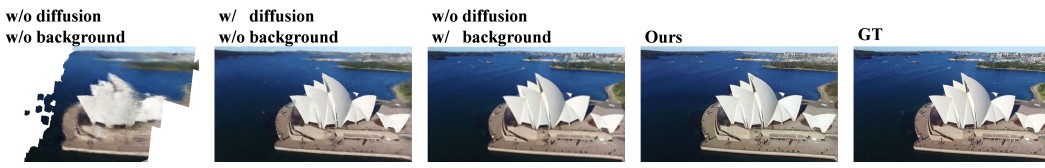

Figure 5: Qualitative performance of ablation experiments. From left to right: removing both the diffusion-based point cloud up-sampling module and the background fusion module, removing only the background fusion module, removing only the diffusion-based point cloud up-sampling module, our PDF method, and the groundtruth.

We also perform ablation experiments to compare our method with other point cloud up-sampling methods. With the same experimental setup, we use a GAN-based method [40] for point cloud up-sampling instead of the diffusion-based up-sampling module. Tab. 3 shows the quantitative results for three scenes (scan5, scan11 and scan12) in the OMMO dataset. Our method exhibits superior performance compared to the GAN-based point cloud up-sampling method, primarily due to its

Table 3: Quantitative results of ablation experiments, including removing the diffusion-based point cloud up-sampling module, using the GAN-based point cloud up-sampling method, our PDF method.

| Method | PSNR↑ | SSIM↑ | LPIPS↓ |
|---|---|---|---|
| w/o diffusion | 21.85 | 0.84 | 0.204 |
| GAN-based method [40] | 24.83 | 0.86 | 0.161 |
| **Ours** | **28.60** | **0.90** | **0.137** |

ability to preserve the structural and topological characteristics of point clouds while effectively handling incomplete or noisy point cloud data. In addition, Fig. 6 shows the visualization results of the diffusion-based point cloud up-sampling module, and our method can not only densify the sparse point cloud reconstructed by the COLMAP, but also fill in the missing regions of the point cloud such as the background and empty space.

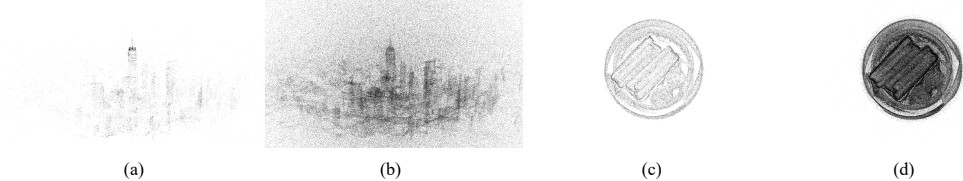

(a)  (b)  (c)  (d)

Figure 6: Qualitative evaluation of the diffusion-based point cloud super-resolution module. From left to right: (a) Point cloud of a large-scale scene reconstructed using COLMAP. (b) Point cloud of the same large-scale scene enhanced using our method. (c) Point cloud of a small-scale scene reconstructed using COLMAP. (d) Point cloud of the same small-scale scene enhanced using our method (zoom-in for the best view).

## 5   Conclusions and Limitations

In this paper, we propose PDF, a point diffusion implicit function for large-scale scene neural representation, and demonstrate its robustness and fidelity on novel view synthesis tasks. The core of our method is to provide dense point cloud surface priors to reduce the huge sampling space of large-scale scenes. Therefore, a point cloud super-resolution module based on diffusion model is proposed to learn from the sparse point cloud surface distribution reconstructed from training views to generate more dense point clouds. However, only constraining the sampling space to the point cloud surface does not fully solve the novel view synthesis problem since point clouds do not have background information. So Mip-NeRF 360 [2] is employed to provide background features and synthesize photo-realistic new perspectives. Extensive experiments demonstrate that our method outperforms current methods in both subjective and objective aspects. At the same time, ablation experiments also prove the effectiveness of our core module, point up-sampling diffusion.

In future work, we will attempt to explore a cross-scene point cloud up-sampling generalization diffusion model instead of training a diffusion model for each scene to improve efficiency. Even more futuristically, it may be possible to extract representative scene representations and inject them into reconstructed point clouds to achieve cross-scene rendering, i.e., generalized point diffusion NeRF.

## Acknowledgements

This work is supported by National Natural Science Foundation of China (No. 62101137, 62071127, and U1909207), Shanghai Natural Science Foundation (No.23ZR1402900), and Zhejiang Lab Project (No.2021KH0AB05).

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
