# OpenReview forum: "PDF: Point Diffusion Implicit Function for Large-scale Scene Neural Representation"
_NeurIPS.cc/2023/Conference — NeurIPS 2023 poster_

### Official Review · Reviewer_v7wN · 2023-07-02

**Soundness:** 3 good
**Presentation:** 3 good
**Contribution:** 3 good
**Rating:** 6
**Confidence:** 5

**Summary:**

This paper proposed a point cloud-based representation for neural rendering of large-scale scenes.  Point cloud diffusion model is designed to upsample the point cloud for better performance. The proposed method achieves state-of-the-art performance in the tested scenes. However, I still have some concerns about the implementation details and the performance comparison.

**Strengths:**

1. The proposed method achieves state-of-the-art performance in the neural rendering of large-scale scenes.
2. Using diffusion models for point cloud upsampling is interesting and useful in the tested scenes.
3. The presentation and writing of the paper is good and the visualization comparison is clear.

**Weaknesses:**

1. Size of the train dataset for diffusion. Diffusion model typically requires large-scale dataset for better performance. I'm curious about the number of training point cloud samples for diffusion model.
2. Reconstruction quality. The 3D reconstruction quality of MVS methods like COLMAP is not guaranteed and there can exist large holes in texture-less regions (e.g., white walls in outdoor scenarios.) I wonder the performance of the diffusion model in such situations (as the diffusion model in the manuscript is mainly designed for upsampling, not completion).
3. Comparison with Point-NeRF. Point-NeRF is an obvious baseline for this method, which should be included in the experiments. The ablations in the experiments can be seen as a variantion of Point-NeRF. However, Point-NeRF include a point grown and pruning strategy, which can also densify the point cloud. It would be better to test the performance of this strategy in large-scale scenes, which has not been explored before.
4. Inference time. The proposed method exploit an explicit reconstruction of the scene, thus the empty space can be simply skipped for better efficiency. However, the complexity of the proposed method is not reported and some implementation details are missed (e.g., the number of sampled points)
5. Background features. Mip-NeRF 360 is utilized for background rendering. However, as shown in Fig.5, the background also contributes to fore-ground regions. It would be better to include an explanation for this.
6. Others. Diffusion models are widely used for generative models and their exsiting randomness in the generated results. However, the 3D reconstruction/the neural rendering should be deterministic. I wonder the influence of the randomness in this method. Besides, I'm also curious about the failure cases of the method.

**Questions:**

Please refer to Weaknesses.

**Limitations:**

Please refer to Weaknesses (especially 2. and 6.).

---

> ### Author Rebuttal · Authors · 2023-08-10
>
> We appreciate your approval of our idea and the detailed and insightful comments. Your concerns will be addressed in the following comments and the final version of our paper will be updated accordingly.
>
> 📝 **Q: Size of the train dataset for diffusion.**
>
> 💡&#8194;**A:**  Our method requires a diffusion model to be trained on each scene, the number of training point cloud samples varies from 40,000 to 100,000 for different scenes.
>
> 📝 **Q: Reconstruction quality.**
>
> 💡&#8194;**A:**  As you have expressed concerns, it is evident that there are missing parts in the point cloud reconstructed by COLMAP, as shown in Figure 9. During the diffusion training process, our training data pairs are obtained by subsampling the reconstructed point cloud twice, transforming a sparser point cloud into a denser one. Since the subsampling rate can approach zero, the training process encompasses cases of completing structural gaps. Therefore, our method is capable of addressing such situations, as depicted in Figure 9.
>
> 📝 **Q: Comparison with Point-NeRF.**
>
> 💡&#8194;**A:** Thank you for your suggestion. Point-NeRF employs a point growing and pruning strategy to generate a surface point cloud sufficient for rendering in small-scale scenes. However, this approach falls short when dealing with sparse and structurally complex point clouds in large-scale scenes. We present the results of Point-NeRF on the OMMO dataset, as shown in the table below.
>
> |Method | PSNR↑ |SSIM↑ |LPIPS↓|
> |:-:|:-:|:-:|:-:|
> |Point-Nerf| 17.85| 0.57 |0.320|
> |**Ours**| **25.10** |**0.79**|**0.205**|
>
> 📝 **Q: Inference time and implementation details.**
>
> 💡&#8194;**A:** Our method avoids ray sampling in the empty scene space by leveraging point clouds, so the inference time will be relatively reduced. We evaluate the training and rendering time of our method on the 5th scene (Sydney Opera House) from the OMMO dataset. At a resolution setting of 1280×676, our method takes about 38 hours to train and 12 seconds to render on a single Nvidia A100 GPU device. Referring to the implementation details of Point-NeRF, we set the number of sampling points on the sampling ray of each pixel to 80, and the features of each sampling point are aggregated from the features of 8 neural points around it.
>
>
> 📝 **Q: Reasons for the facilitative impact of the background on the foreground.**
>
> 💡&#8194;**A:**  During the training process, the foreground and background are jointly optimized. If the background exhibits poor performance, it can lead to significant losses and adversely affect the optimization of the foreground.
>
>
> 📝 **Q: the influence of the randomness.**
>
> 💡&#8194;**A:**  In order to explore the influence of this randomness on our method, we conducted experiments on three scenes(scan5, scan11, scan12) from the OMMO dataset. We set three different random number seeds for each scene, and the quantitative results are shown in  the table below. The metrics show that the impact of different random number seeds on rendering quality is almost negligible. In addition, our method has no failure cases on the OMMO dataset,  Figure 10 in the attached PDF shows the rendering results of all scenes.
>
> |Seed | PSNR↑ |SSIM↑ |LPIPS↓|
> |:-:|:-:|:-:|:-:|
> |2| 28.60 | **0.90** |**0.136**|
> |4| **28.63** | **0.90** |**0.136**|
> |8| 28.42 | 0.89 |0.140 |
> |**Ours**| 28.60 |**0.90** |0.137|

---

> > ### Comment · Reviewer_v7wN · 2023-08-15
> > **Response**
> >
> > I appreciate the detailed rebuttal.
> > As for the performance of Point-NeRF, have you also used background fusion for Point-NeRF (maybe a simple fusion with MipNeRF 360)? I observed that the performance of Point-NeRF is even worse than the version of yours w/o diffusion (according to the rebuttal for Reviewer yNkC).

---

> > > ### Author Response · Authors · 2023-08-15
> > > **Response to the Performance of Point-NeRF**
> > >
> > > Thank you for your insightful comments.
> > >
> > > We sincerely apologize for the confusion. We present our evaluation results of Point-NeRF on the entire OMMO dataset and provide ablation experiments for Reviewer yNkC (due to time and computational constraints, all ablation experiments are evaluated on a single scene, the Sydney Opera House).
> > >
> > > Since Point-NeRF can only reconstruct sparse foreground point clouds, it cannot be directly applied to large-scale outdoor scenes. The evaluation results on the entire OMMO dataset are as follows:
> > >
> > > |Method | PSNR↑ |SSIM↑ |LPIPS↓|
> > > |:-:|:-:|:-:|:-:|
> > > |Point-NeRF| 17.85| 0.57 |0.320|
> > > |**Ours**| **25.10** |**0.79**|**0.205**|
> > >
> > > Furthermore, we provide a set of ablation studies to analyze the performance gains brought by our point super-resolution diffusion module and the background module. Among them, "w/o diffusion, w/o background" is the Point-NeRF method, and "w/o diffusion, w/ background" is Point-NeRF with background fusion. It can be observed that, compared to Point-NeRF, our point super-resolution diffusion module can bring significant gains by providing a dense surface point cloud (Point-NeRF vs. Our foreground).
> > >
> > > |Method | Known as |PSNR↑ |SSIM↑ |LPIPS↓|
> > > |:-:|:-:|:-:|:-:|:-:|
> > > |w/o diffusion, w/o background| Point-NeRF |9.28| 0.51 |0.355|
> > > |w/o diffusion, w/ background| Point-NeRF + background |21.05| 0.83 |0.219|
> > > |w/ diffusion, w/o background| Our foreground  |22.93| 0.78 |0.235|
> > > |**w/ diffusion, w/ background**|**Ours**|**27.58** |**0.90**|**0.162**|
> > >
> > > In the final manuscript, two tables will be included separately as Table 1 in the Performance Comparison section and as Table 3 in the Ablation Studies section, accompanied by detailed explanations to avoid confusion.

---

> > > > ### Comment · Reviewer_v7wN · 2023-08-16
> > > > **Response**
> > > >
> > > > Thanks.
> > > > Basically I think the idea by using diffusion model to enhance point cloud for better rendering quality is insightful. Thus, I will keep my original score (Weak Accept).

---

> > > > > ### Author Response · Authors · 2023-08-20
> > > > >
> > > > > We appreciate your valuable and insightful comments. We feel glad about your generally favorable assessment of our method. Additional evaluation/ablation and corresponding explanations will be included in the final version.

---

### Official Review · Reviewer_H9co · 2023-07-03

**Soundness:** 3 good
**Presentation:** 3 good
**Contribution:** 3 good
**Rating:** 6
**Confidence:** 4

**Summary:**

The paper presents a method for reconstructing large-scale scenes from multi-view images. Given the sparse point cloud computed from an MVS pipeline, the algorithm first employs a point cloud diffusion model to upsample the point cloud. The utilization of the diffusion model could effectively create a dense cloud with missing regions completed. Then, a Point-NeRF-style model is used to reconstruct the scene based on the upsampled point. Foreground and background renderings are fused via an additional fusion network, resulting in the final output. The algorithm is tested on OMMO and BlendMVS datasets. Compared to existing NeRF baselines, the proposed pipeline could limit the sampling region to the surface and achieves better reconstruction and rendering quality.

**Strengths:**

1. Using the diffusion model for point cloud upsampling and completion is novel. While previous methods perform point cloud upsampling or completion using VAE or GAN models, this is the first paper that leverages the power of diffusion models. One related work could be Luo et al. [a], yet the proposed algorithm treats the partial input cloud as a condition and hence is also able to generate an arbitrary number of points.
2. The results are compelling. Using the upsampled point cloud as the supporting domain of NeRF, the method could reconstruct a much larger scene with finer details. Both the qualitative and quantitative experiments demonstrate the effectiveness of the proposed approach.

[a] Luo, Shitong, and Wei Hu. "Diffusion probabilistic models for 3d point cloud generation." Proceedings of the IEEE/CVF Conference on Computer Vision and Pattern Recognition. 2021.

**Weaknesses:**

1. While the use of the diffusion model for point cloud upsampling is novel, its combination with the subsequent NeRF model is very straightforward and detached. The NeRF model is only a slight modification from Point-NeRF, with the addition of the background MLP, and such a representation itself is not insightful enough to form an individual contribution.
2. Some descriptions regarding the method section and experiment section are not very clear. What is the training scheme of the diffusion model? (more details in the 'Question' section). What does '... as a feature $f_i$ of each sampling point $p_i$ to equip structural information' in Line 172 mean?
3. Ablation study using other point cloud upsampling methods is missing. Given the main contribution of the paper is the diffusion-based upsampling network, it is worth investigating if a diffusion-based backbone could surpass a traditional GAN-based backbone. The study could be conducted on a simple toy dataset.

**Questions:**

The main question arises from the training strategy as well as the model specifications of the diffusion model:
1. What is the model architecture of the denoiser? How many parameters are needed?
2. Which dataset is used to train the diffusion model? How many training samples are there? Is data augmentation applied?
3. How many points are generated from the diffusion model?
4. What is the mechanism that allows the diffusion model to complete missing regions (as shown in Fig. 5)? Are regions randomly masked out during the training stage to encourage the model to hallucinate missing contents?

**Limitations:**

Not applicable.

---

> ### Author Rebuttal · Authors · 2023-08-10
>
> We appreciate your approval of our idea and the detailed and insightful comments. Your concerns will be addressed in the following comments and the final version of our paper will be updated accordingly.
>
> 📝 **Q: The combination of novel diffusion-based point cloud upsampling models and Point-NeRF-based NeRF models is relatively weak.**
>
> 💡&#8194;**A:** For unbounded, large-scale outdoor scenes, Point-NeRF-based NeRF models are unable to accurately represent and synthesize fine-grained textures due to the vast sampling space and excessively sparse point clouds, as seen in the first column of Figure 5. To overcome this critical issue, we propose a novel diffusion-based point cloud upsampling module that generates dense scene surfaces. By incorporating an explicit surface prior, we reduce the sampling space of the Point-NeRF-based NeRF model from an unbounded 3D urban-level scale to the scene surface. This effective combination of explicit and implicit representations is the key to the success of our method and provides an effective solution for large outdoor scenes. Furthermore, the effectiveness of large-scale point cloud upsampling is difficult to measure through visualization due to the high degree of object occlusion and overlap. Neural rendering provides a solution for evaluating this research.
>
> 📝 **Q: GAN-based point cloud upsampling approaches.**
>
> 💡&#8194;**A:**  We appreciate your insightful suggestion. In conjunction with the recommendation from Reviewer H9co, we have replaced our diffusion-based method with a Generative Adversarial Network (GAN)-based point cloud up-sampling approach[1].
>
> 📝 **Q: What is the training scheme of the diffusion model?**
>
> 💡&#8194;**A:**  The structure of our diffusion model is referenced to PVD[2], which has a number of parameters of about 27.6 M. We use the sparse point cloud reconstructed by COLMAP as input to train the diffusion model. Our method requires a diffusion model to be trained on each scene, the number of training point cloud samples varies from 40,000 to 100,000 for different scenes. We didn't use data augmentation. The diffusion model generates 1848 points per round based on 200 prior points, and a total of 200 rounds are performed, i.e., 369600 points are generated. For missing regions completions specifically, the model takes as input 200-point partial points and 1,848 points sampled from noise, totaling 2048 points. At each step, the first 200 of the 2,048 points sampled by the model are replaced with the input partial points. The updated point set is then used as input in the next time step. We didn't use the random masking region strategy.
>
> [1] Junzhe Zhang, Xinyi Chen, Zhongang Cai, Liang Pan, Haiyu Zhao, Shuai Yi, Chai Kiat Yeo, Bo Dai, and Chen Change Loy. Unsupervised 3d shape completion through gan inversion. In CVPR, 2021.
>
> [2] Linqi Zhou, Yilun Du, and Jiajun Wu. 3d shape generation and completion through point-voxel diffusion. In Proceedings of the IEEE/CVF International Conference on Computer Vision (ICCV), pages 5826–5835,October 2021.

---

> > ### Comment · Reviewer_H9co · 2023-08-20
> >
> > Dear authors,
> >
> > Sorry for the late reply as I've been on a very busy schedule recently. You just said that your diffusion model is trained for EACH scene (independently?) which added to my confusion. If I understand correctly, essentially you are training the diffusion in a self-supervised manner, so even if you don't have ground truths for a denser point cloud, you could still train from the sparse COLMAP points.
> >
> > This, however, is different from the GAN inversion method [1] where a prior is already learned through a training set. Hence I am wondering if the comparison is still fair in this case.
> >
> > Best,

---

> > > ### Author Response · Authors · 2023-08-20
> > >
> > > Thank you for taking time out of your busy schedule to make insightful comments.
> > >
> > > For each scene, we train one diffusion model. We downsample the sparse point cloud x_s reconstructed by COLMAP to get an even sparser point cloud z_0. Then we further downsample z_0 to get the sparsest point cloud x_0, where x_s, z_0 and x_0 have progressively sparser relationships. Our training process recovers z_0 from the sparsest x_0. During testing, we take x_s as input to generate a denser super-resolved point cloud. This learns the prior for each scene. When using GAN inversion method for point cloud upsampling, we employ the same data organization and augmentation. So, our method is somewhat relatively fair to the GAN inversion method.
> > >
> > > At the same time, we would like to explain the necessity of independent training. The distribution of point clouds varies greatly from scene to scene, especially for large-scale scenes. For this a single pre-trained model has limited generalization capability, while independent training can better capture unique geometries of the scene. Moreover, our diffusion model is still somewhat robust. The network architectures and hyperparameters are shared without re-designing. Only the weights differ.

---

> > > > ### Comment · Reviewer_H9co · 2023-08-20
> > > >
> > > > Dear authors,
> > > >
> > > > Your newly described method now makes sense to me, and please make sure to incorporate them into your revision. Basically, you're leveraging the internal regularity of the neural network itself to upsample (or more precisely, 'interpolate') COLMAP's sparse points, with no intention of making it generalizable to other scenes since you fit a single network for each one.
> > > >
> > > > I've now understood all the necessary settings of the paper and I've raised my rating from weak reject to weak accept. Hope this will get your paper through. Thanks!
> > > >
> > > > Best,

---

> > > > > ### Author Response · Authors · 2023-08-20
> > > > >
> > > > > We appreciate your valuable and insightful comments. We feel glad about your generally favorable assessment of our method. Additional evaluation/ablation and corresponding explanations will be included in the final version.

---

### Official Review · Reviewer_rzgR · 2023-07-06

**Soundness:** 2 fair
**Presentation:** 2 fair
**Contribution:** 1 poor
**Rating:** 4
**Confidence:** 4

**Summary:**

This paper proposes an implicit neural representation for large-scale scenes with two major components, the first one is the point diffusion implicit function (PDF) which adopts diffusion process to generate dense point cloud from point cloud produced by Colmap, the second one is the background rendering which basically borrow the idea from Mip-Nerf360.

**Strengths:**

•	The paper is well organized and easy to follow.
•	The qualitative comparison on the fly-view dataset clearly shows the effectiveness of the proposed method on the large-scale scenes, where the rendered images are sharp in both foreground and background.
•	Most of nerf paper focus on small or medium scene, this paper instead provides a interesting solution to the large scale scene representation.

**Weaknesses:**

•	The effectiveness of the diffusion process, the motivation behind adding the diffusion process over colmap generated point cloud is not clear to me since the PointNerf method can generate clean and sharp foreground rendering directly with the colmap point cloud. Some explanation or visualization showing the point cloud before and after diffusion will be very helpful.
•	One major advantages of point-cloud based neural field is they can avoid prohibitive reconstruction time of Nerf, so a question what is the training time and rendering speed of this paper. Will the diffusion process significantly slow down the training speed?
•	The overall technical contribution of this paper is weak. The main component PDF shares the major design with PointNerf except applying a diffusion process over the Colmap-generated point cloud. Currently there is no enough discussion over the insight behind this design as I said before. The background rendering module also follows the existing technique from Mip-Nerf 360. Therefore I think the contribution is not enough.
•	In the experiment part the author said they use two dataset for evaluation but I didn’t find the result on BlendMVS


**Questions:**

•	How was the diffusion process necessary for large-scale scene representation, I do see an ablation study about that module, but I am not sure why removing the diffusion make the method totally failed but the PointNerf can works well with pointcloud directly from COLMAP. More explanation or visualization will be very helpful.
•	What’s the runtime of the proposed methods.
•	A general (but maybe not that related to the proposed method): Is there any other solutions to the background rendering? Will the environment map work for the background of the large outdoor scene.


**Limitations:**

The authors have discussed potential negative social impacts of this paper.

---

> ### Author Rebuttal · Authors · 2023-08-10
>
> We appreciate your approval of our idea and the detailed and insightful comments. Your concerns will be addressed in the following comments and the final version of our paper will be updated accordingly.
>
> 📝 **Q: The combination of novel diffusion-based point cloud upsampling models and Point-NeRF-based NeRF models is relatively weak.**
>
> 💡&#8194;**A:** For unbounded, large-scale outdoor scenes, Point-NeRF-based NeRF models are unable to accurately represent and synthesize fine-grained textures due to the vast sampling space and excessively sparse point clouds, as seen in the first column of Figure 5. To overcome this critical issue, we propose a novel diffusion-based point cloud upsampling module that generates dense scene surfaces. By incorporating an explicit surface prior, we reduce the sampling space of the Point-NeRF-based NeRF model from an unbounded 3D urban-level scale to the scene surface. This effective combination of explicit and implicit representations is the key to the success of our method and provides an effective solution for large outdoor scenes. Furthermore, the effectiveness of large-scale point cloud upsampling is difficult to measure through visualization due to the high degree of object occlusion and overlap. Neural rendering provides a solution for evaluating this research.
>
> 📝 **Q: The results on the BlendMVS dataset.**
>
> 💡&#8194;**A:** We sincerely apologize for the confusion. Due to limitations in space and time, we have included the experimental results of the BlendMVS dataset in the supplementary materials. For further details, please refer to Table 4 and Figure 6 in the supplementary materials, which will be incorporated into the final version of the manuscript.
>
> 📝 **Q: The necessity of the diffusion module.**
>
> 💡&#8194;**A:** For outdoor urban-level scenes, the point cloud reconstructed using COLMAP tends to be sparse, especially in unbounded background regions, resulting in blurry foreground and missing background in rendered images. To demonstrate this, we provide quantitative results of applying Point-Nerf to the unbounded large-scale OMMO dataset in the following table. It can be observed that Point-Nerf is not directly applicable to large scenes. Additionally, we present visualization results as shown in Figure 9, where severe occlusions in outdoor large-scale scenes cause the loss of most geometric details when projecting the point cloud onto 2D. To facilitate performance observation, we conducted experiments on the hotdog scene in the NeRF-Synthetic dataset. It can be observed that our method not only enhances the geometric structure of the foreground but also fills in the missing parts.
>
> |Method | PSNR↑ |SSIM↑ |LPIPS↓|
> |:-:|:-:|:-:|:-:|
> |Point-Nerf| 17.85| 0.57 |0.320|
> |**Ours**| **25.10** |**0.79**|**0.205**|
>
> 📝 **Q: What's the runtime of the proposed methods.**
>
> 💡&#8194;**A:** We evaluate the training and rendering time of our method on the 5th scene (Sydney Opera House) from the OMMO dataset. At a resolution setting of 1280×676, our method takes about 38 hours to train and 12 seconds to render on a single Nvidia A100 GPU device.
>
> 📝 **Q: Additional solutions for background rendering.**
>
> 💡&#8194;**A:** Foreground-background modeling, originating from NeRF++, is a commonly used method for scene neural implicit representation. This approach includes subsequent works such as Neurs, VolSDF, CoCo-INR, among others. Additionally, environmental texturing is a prevalent technique, often applied to model individual objects. However, for scenes, the background is often an integral component, making the use of foreground-background modeling more prevalent.

---

### Official Review · Reviewer_DZNn · 2023-07-07

**Soundness:** 3 good
**Presentation:** 3 good
**Contribution:** 3 good
**Rating:** 6
**Confidence:** 4

**Summary:**

In this paper, the authors propose a new approach to tackle the reconstruction of Neural Radiance Fields from large, unbounded scenes. A major limitation of current neural fields is the lack of scalability due to the number of sampling points in empty space, which is limiting the achievable scale or quality with limited compute resources. In this method, the sampling space is restricted to the surface of the object. To get an approximation of the surface of the scene, they utilize a point cloud, which first is reconstructed through MVS and upscaled to achieve a dense point cloud on the surface using a diffusion probabilistic model. Only sampling points in the near distance of the surface point cloud and the respective feature, which is constructed as a predicted feature from the neighborhood points and sampling points and forms the foreground feature together with the constructed feature. The unbounded background of the scene is modeled on a sphere as features. Foreground and background features are fused in another MLP.

**Strengths:**

- The presented method builds on known techniques for neural rendering and generative super-resolution and combines those to solve a novel task of reconstructing large-scale unbounded scenes. I think this is a big strength of the paper, which is justified by qualitative and quantitative results and ablation studies.
- The training approach shows a clear explanation of how the missing ground truth data for the point cloud can be optimized with the limited resources
- The authors present their method in an understandable way, adding the necessary context and references to reproduce the work
- Presented results show a clear advantage over standard NeRF techniques, and better results in areas of fine textures compared to methods designed for large-scale or unbounded scene
- Most design choices are ablated and justified in separate studies

**Weaknesses:**

- One major weakness of the proposed method is that it is required to retrain the whole model per scene. This is also mentioned by the authors in their limitations but leads to really long training times compared to sota methods.
- While the result section presents all relevant quantitative results in a table, the authors do not provide an explanation for some of the outliers and rather good performances of the baselines in some scenes.
- I hope I have not missed something, but there is no explanation of Fig.4. To my understanding, this provides a scene in which Mip-NeRF 360 fails completely to reconstruct the scene, but no explanation is given. Also, I am curious why it results in something like this and not only blurry results as reported in Fig.3
- One major limitation or experiment important for the specific task of large-scale, outdoor scene reconstruction is to show the boundaries of the method in terms of scene scale. Mega-NeRF is specially designed to divide a scene in multiple sections and might be able to handle larger scenes, which is not the case for this method. An additional study on the limitation of this method in terms of scene size alongside with training and rendering times would probably just strengthen the claims

**Questions:**

- Can you provide the training and rendering times for a given resolution for your and all reference methods
- The authors mentioned that the method was only evaluated on a subset provided by the authors of OMMO, but the full dataset results in the supplementary provide similar insights.

**Limitations:**

Most limitations are mentioned at the end of the paper. As discussed in the weakness section, I think an evaluation of limitation wrt scene size and computational times in a similar setting, would provide a better understanding of the capabilities of this methos.

---

> ### Author Rebuttal · Authors · 2023-08-10
>
> We appreciate your approval of our idea and the detailed and insightful comments. Your concerns will be addressed in the following comments and the final version of our paper will be updated accordingly.
>
> 📝 **Q: long training times caused by per-scene optimization.**
>
> 💡&#8194;**A:** Due to the challenge of establishing effective and generalizable representations for large-scale scenes containing a multitude of object categories, the common practice is to optimize on a per-scene basis. In comparison to existing state-of-the-art (SOTA) methods, our rendering time is acceptable, thanks to the utilization of a dense point cloud prior that reduces rendering time. A comparison of training times with SOTA methods is presented in the table below and will be included in our final version. Additionally, in future work, we aim to explore the utilization of a cross-scene point cloud upsampling generalization using a diffusion model, instead of training a diffusion model for each scene, to enhance efficiency.
> |Method | Nerf |Nerf++ |Mip-Nerf|Mip-Nerf 360|Point-Nerf|Ours|
> |:-:|:-:|:-:|:-:|:-:|:-:|:-:|
> |Training time(h)| 7.2 | 9.5 |11.2|9.1|8.0|38.0|
> |Tendering time(s)| 3.4 | 5.2 |8.7|10.3|6.3|12.0|
>
>
> 📝 **Q: Explanation for some of the outliers and performances.**
>
> 💡&#8194;**A:** Thank you for your suggestion. Our proposed method does not have any bad cases, but it may exhibit suboptimal performance in scenes with relatively flat spatial structures, such as grasslands (scan6, scan17), roads (scan1), and plazas (scan9). This is because the reconstructed point cloud is approximately planar and lacks sufficient geometric priors present in other scenes. Similarly, NeRF tends to fail in unbounded scenes (scan1, scan27, scan33) and scenes with abundant reflective surfaces (scan26), which is similar to Mip-NeRF (scan22, scan32, scan33, scan26). NeRF++, Mega-NeRF, and Ref-NeRF are relatively robust methods, but they may produce relatively blurry renderings for large-scale urban scenes. Mip-NeRF 360 requires high data robustness, as less than 10% of abnormal camera poses can lead to the failure of the entire scene (scan16).
>
> 📝 **Q: An analysis of the causes of the bad case in Mip-NeRF 360.**
>
> 💡&#8194;**A:** As illustrated in Figure 4, mipnerf360 encountered failure in an outdoor jungle scene characterized by a substantial amount of repetitive textures and similar details. We visualized the camera trajectory provided by the scene and cross-referenced it with the original video, revealing an erroneous calibration of a specific viewpoint (as depicted in Figure 12 in the attached PDF), which likely had a significant impact on the performance of Mip-NeRF 360. In contrast, our approach exhibits enhanced robustness by reconstructing and enhancing the scene point cloud, providing a visual representation that incorporates prior knowledge to counteract partially erroneous data.
>
> 📝 **Q: The method's upper limit on scene scale.**
>
> 💡&#8194;**A:** Thank you for your suggestions. According to the OMMO dataset, their scene areas range from 2km2 to 1000 km2. Mega-NeRF divides a large scene into multiple smaller scenes to accelerate and reduce the fitting difficulty of each NeRF unit. However, for city-scale scenes, each subpart remains too large to generate highly detailed textures. We did not adopt this approach, considering that partitioning would disrupt the integrity of the geometric structure, which is unfavorable for the point cloud super-resolution module to learn the underlying structure of the entire scene.
>
> 📝 **Q: dataset issues.**
>
> 💡&#8194;**A:** We sincerely apologize for the confusion. Due to the initial release of only a representative subset of the OMMO dataset, we obtained the complete data shortly before the submission deadline. As a result, the additional scene performance was included in the supplementary materials. In the final version, these two parts will be merged to report the overall results on the entire OMMO dataset.

---

> > ### Comment · Reviewer_DZNn · 2023-08-17
> > **Commet**
> >
> > Thank you for the extensive answers to my questions. Those provide the necessary context to understand the results and computational costs of the method and I encourage the authors to add those to the main paper
> >
> > Wrt. the bad case in Mip-NeRF360 I do not think it is a fair comparison for this specific case. Although Figure 4 is presenting robustness across scenes. With the lack of context provided in the attached pdf, this is not painting representative of the performance of the method. It would be good to either change the wording or add some context to the Figure description.
> >
> > Thank you for addressing the confusion concerning the OMMO dataset. That makes sense!

---

> > > ### Author Response · Authors · 2023-08-18
> > >
> > > We appreciate your constructive feedback and are pleased to address the majority of your concerns.
> > >
> > > When considering the performance comparison with Mip-NeRF 360, since they are all based on the same data, all evaluation methods exhibit a degree of relative comparability, even if individual scenes have incorrectly calibrated perspectives. Concurrently, large-scale NeRF often necessitates enhanced robustness. Given that real outdoor large-scale scenes inherently encompass numerous disturbances that are unfavorable to NeRF, such as moving individuals or vehicles, alterations in illumination, high dynamic range, etc., inaccurately calibrated viewpoints can, in a sense, be regarded as a form of disturbance.
> > >
> > > Furthermore, the table below presents experiment results excluding failed scenes for Mip-NeRF 360, where our PDF method still outperforms Mip-NeRF 360.
> > >
> > > |Method | PSNR↑ |SSIM↑ |LPIPS↓|
> > > |:-:|:-:|:-:|:-:|
> > > |Mip-NeRF 360| 23.85 | 0.70 |0.395|
> > > |**Ours**| **25.30** |**0.80**|**0.199**|
> > >
> > > We are appreciative of your constructive feedback, which significantly contributes to the enhancement of our manuscript's quality. Additional evaluations/ablations and corresponding elucidations will be incorporated into the final version.

---

> > > > ### Comment · Reviewer_DZNn · 2023-08-21
> > > >
> > > > Thank you for your prompt response, I really appreciate the additional results and explanation. I agree with the authors, that all methods are  exposed to incorrect calibrations. And the results show that the proposed method does perform better independent of failure cases in other methods. But I still think, that a more thorough explanation needs to be added to all figures and results in the main manuscript, otherwise those lack context.
> > > >
> > > > Due to some limitations, such as long training times per scene and scene size, and after carefully reading all other discussions I think my rating is still justified.

---

> > > > > ### Author Response · Authors · 2023-08-21
> > > > >
> > > > > We appreciate your valuable and insightful comments. We feel glad about your generally favorable assessment of our method. Additional evaluation/ablation and corresponding explanations will be included in the final version.

---

### Official Review · Reviewer_yNkC · 2023-07-08

**Soundness:** 2 fair
**Presentation:** 3 good
**Contribution:** 2 fair
**Rating:** 4
**Confidence:** 4

**Summary:**

The paper proposed a novel method for the novel view synthesis of large-scale outdoor scenes. The method combines a Point-NeRF-baed rendering stage for the foreground, as well as a background stage based on Mip-NeRF 360. To cope with the issue of overly sparse point cloud from COLMAP MVS, a point could diffusion model is trained on each individual scene from existing MVS point cloud, and used to synthesize more points. The proposed method achieves leading performance on both large and small scenes.

**Strengths:**

* Training a scene-specific diffusion model for point cloud upsampling is a novel approach that seems to work well.
* The proposed method is able to achieve better quality on a large set of real-world scenes both quantitatively and qualitatively.

**Weaknesses:**

* The proposed method requires a diffusion model to be trained on each scene, which can require significant computation.
* In terms of comparison with previous works, it is not sure if the comparison is based on a level ground, as the proposed method might require more computation, more parameter count, or both.
* It is not sure if the proposed method is able to outperform methods that are based on grid data structure, such as InstantNGP, Plenoxels, etc., which tend to be very fast and scalable to large, sparse and unbounded scenes.
* It is not sure if other ways of upsampling the point clouds will also work. Although the use of a scene-specific diffusion model to upsample point could is novel, it is not compared against any baselines. It might provide better insight and more helpful for future work if this contribution can be studied independently as a separate work.

**Questions:**

* Could you elaborate more on the reason behind the failure of Mip-NeRF 360 in Figure 4?
* Why is the background partially missing in Fig. 5 column 1?

**Limitations:**

Although there is a Limitations section, it only includes future works and there lacked discussions on the limitations.

---

> ### Author Rebuttal · Authors · 2023-08-10
>
> We appreciate your approval of our idea and the detailed and insightful comments. Your concerns will be addressed in the following comments and the final version of our paper will be updated accordingly.
>
> 📝 **Q: The significant computation caused by per-scene optimization.**
>
> 💡&#8194;**A:** As your concern, pre-scene optimization is currently a prevalent approach for constructing implicit neural representations. Although there have been some advancements in NeRF generalization methods, they are still limited to representing toy or mini-abstract scenes. This limitation arises from the inherent difficulty in establishing a universal and effective representation for complex scenes or objects, particularly for unbounded outdoor scenes that encompass thousands of objects across numerous categories within each scene. Consequently, this paper adopts the per-scene optimization strategy to construct dense point clouds and neural implicit fields for each scene, thereby facilitating the simultaneous learning of representations for multiple object categories within the scene.
>
> 📝 **Q: Performance improvement due to large model size or design.**
>
> 💡&#8194;**A:**  To validate the improvement originating from our proposed point cloud upsampling module versus additional parameters, we expanded the MLP layers of PointNeRF and Mip-NeRF360 to roughly match the parameter count of our approach. The following table shows the experimental results. Directly increasing the parameter count does not lead to additional performance improvements, as it can make the network more challenging to optimize and prone to overfitting on the training viewpoints (Due to time constraints, this experiment is exclusively validated on scan5).
> |Method |Params | PSNR↑ |SSIM↑ |LPIPS↓|
> |:-:|:-:|:-:|:-:|:-:|
> |Enlarge PointNeRF| 38.19M |19.92| 0.77 |0.362|
> |Enlarge Mip-NeRF360| 36.10M |22.01| 0.77 |0.322|
> |**Ours**| 38.68M| **27.58** |**0.90** |**0.162**|
>
> 📝 **Q: Additional point cloud upsampling approaches.**
>
> 💡&#8194;**A:**  We appreciate your insightful suggestion. In conjunction with the recommendation from Reviewer H9co, we have replaced our diffusion-based method with a Generative Adversarial Network (GAN)-based point cloud up-sampling approach[1].  As evident from the table below, our method exhibits superior performance compared to the GAN-based point cloud upsampling method, primarily due to its ability to preserve the structural and topological characteristics of point clouds while effectively handling incomplete or noisy point cloud data (Due to time constraints, this experiment is exclusively validated on scan5, scan11 and scan12).
> |Method | PSNR↑ |SSIM↑ |LPIPS↓|
> |:-:|:-:|:-:|:-:|
> |GAN-based method| 24.83| 0.86 |0.161|
> |**Ours**| **28.60** |**0.90** |**0.137**|
>
>
> 📝 **Q: An analysis of the causes of the bad case in Mip-NeRF 360.**
>
> 💡&#8194;**A:** As illustrated in Figure 4, MipNeRF 360 encountered failure in an outdoor jungle scene characterized by a substantial amount of repetitive textures and similar details. We visualized the camera trajectory provided by the scene and cross-referenced it with the original video, revealing an erroneous calibration of a specific viewpoint (as depicted in Figure 12 in the attached PDF), which likely had a significant impact on the performance of Mip-NeRF 360. In contrast, our approach exhibits enhanced robustness by reconstructing and enhancing the scene point cloud, providing a visual representation that incorporates prior knowledge to counteract partially erroneous data.
>
> 📝 **Q: The reasons for the background missing when removing the diffusion-based point cloud up-sampling module.**
>
> 💡&#8194;**A:** We apologize for any confusion. Our intention with the ablation study was to create an additive chain, starting from the original multi-view reconstruction point cloud and progressively incorporating our diffusion super-resolution module and background module. Therefore, 'w/o diffusion' actually implies 'w/o diffusion and background'. In our work with city-scale outdoor scenes, the point clouds obtained from multi-view reconstruction methods are extremely sparse, particularly in boundary-less background areas, leading to the absence of background in the rendered images. We will rectify this in the final version and update the results of the 'w/o diffusion' experiment(Due to time constraints, this experiment is exclusively validated on scan5).
>
> |Method | PSNR↑ |SSIM↑ |LPIPS↓|
> |:-:|:-:|:-:|:-:|
> |w/o diffusion, w/ background| 21.05| 0.83 |0.219|
> |**Ours**| **27.58** |**0.90**|**0.162**|
>
> 📝 **Q: Comparison with methods based on grid data.**
>
> 💡&#8194;**A:** In contrast to nerf, InstantNGP uses a sparse parameterized voxel grid instead of mlp for scene representation, which compresses the training time from hours to minutes or even seconds. To explore the performance of this method on unbounded large-scale scenes, we trained InstantNGP on three scenes from the OMMO dataset. With a resolution setting of 640*320, after 10 seconds of training, the PSNR of the rendered images averages about 25.69. Qualitative results(as depicted in Figure 11 in the attached PDF) show that although InstantNGP can perform 3D scene modeling very quickly, it is hard to render the details of foreground objects and the background is often completely blurred.
>
> [1] Junzhe Zhang, Xinyi Chen, Zhongang Cai, Liang Pan, Haiyu Zhao, Shuai Yi, Chai Kiat Yeo, Bo Dai, and Chen Change Loy. Unsupervised 3d shape completion through gan inversion. In CVPR, 2021.

---

> > ### Comment · Reviewer_yNkC · 2023-08-14
> > **Further questions**
> >
> > I would like to thank the authors for the rebuttal. Some further questions:
> > 1. For the InstantNGP experiment, will the result improve with longer training? 10 seconds of training seems to be too short.
> > 2. I wonder if you have used the NeRF++/MipNeRF360 scene contraction in the Instant NGP experiment?

---

> > > ### Author Response · Authors · 2023-08-14
> > >
> > > Thank you for your insightful comments.
> > >
> > > Regarding the first issue, on the OMMO dataset, InstantNGP demonstrates the ability to converge within a short training time and achieve comparable reconstruction quality. Further increasing the training time does not result in significant improvements in quality, aligning with the original design intention of InstantNGP as a fast-converging framework.
> > >
> > > Regarding the second issue, we do not use scene contraction in the InstantNGP. This is because scene contraction requires applying spatial transformations to compress the scene to the unit sphere, which requires modifying the ray-marching logic to match the curvature of the rays to the transformations, increasing the complexity of the algorithm. The spatial transformation also affects the resolution distribution of the encoding; the encoding in InstantNGP depends on linear resolution, and scene contraction breaks the resolution linear relationship, which reduces the reconstruction quality.

---

> > > > ### Comment · Reviewer_yNkC · 2023-08-18
> > > >
> > > > Thanks for the further clarification. I am inclined to keep my original rating of Borderline Reject, leaning towards reject. First, without comprehensive comparison on different datasets and different parameter tuning, the comparison with InstantNGP is likely unreliable. Further, even if there is performance gain, it is difficult to justify the proposed method given its significantly longer training time (days vs 10 seconds) and more complicated training procedure.

---

> > > > > ### Author Response · Authors · 2023-08-21
> > > > >
> > > > > Thank you for your additional comments and suggestions.
> > > > >
> > > > > OMMO dataset is a real outdoor large-scale dataset that perfectly aligns with the settings our method aims to address. Additionally, we have validated our method on five synthetic large-scale scenes from the BlendedMVS dataset. Extensive experiments have demonstrated the effectiveness of our method for large-scale scene novel view synthesis, both in terms of quantitative and qualitative evaluations. Regarding parameter tuning, a uniform set of parameters is employed across all scenes, as mentioned in Section 3.3 - Implementation Details. A comprehensive evaluation of parameter ablation will be added to our final manuscript.
> > > > >
> > > > > Furthermore, long-time training PSNR evaluation results for InstantNGP are provided. It can be observed that prolonged training does not yield significant benefits for it (due to rebuttal time constraints, the ablation experiments on the OMMO dataset were only validated on scan5, scan11, and scan12).
> > > > >
> > > > > |Method |dataset| 10s | 1min |10hour|
> > > > > |:-:|:-:|:-:|:-:|:-:|
> > > > > |InstantNGP|OMMO|25.69|25.74|26.32|
> > > > > |**Ours**|OMMO| 11.31|13.42|**27.58**|
> > > > > | InstantNGP |BlendedMVS|17.02|17.19|17.63|
> > > > > |**Ours**| BlendedMVS| 10.44|12.98|**20.21**|
> > > > >
> > > > > It is worth noting that the design objectives and application scenarios of InstantNGP differ somewhat from ours. It focuses more on real-time rendering and interaction, while we prioritize fine-grained large-scale scene representation and high-quality rendering. Therefore, compared to InstantNGP, our method reconstructs detailed textures more finely, and the lighting effects are more realistic (cf. Fig 11). We believe that achieving a qualitative leap within a certain time cost is meaningful.
> > > > >
> > > > > We greatly appreciate your suggestions, as they immensely contribute to enhancing the quality of our paper. We hope that our response can address your concerns and gain your approval.

---

### Author Rebuttal · Authors · 2023-08-10

We thank all reviewers for their constructive feedback and recognizing our method as **achieving better qualitative and quantitative performance** (yNkC, DZNn, rzgR, H9co, v7wN); **introducing a novel and effective diffusion-based up-sampling module for large-scale point clouds** (yNkC, DZNn, H9co, v7wN); **capable of generating detailed texture areas** (DZNn, rzgR, H9co, v7wN); and **being well-organized and easy to understand** (DZNn, rzgR, v7wN). We will address their concerns below.

---

### Decision · Program_Chairs · 2023-09-21

**Decision:**

Accept (poster)

**Comment:**

NVS method. Innovation is per-scene diffusion trained on points used for inpainting.

Downside is, that the comparison is not as substantiated as several R would like it to be. Something much simpler for upsampling might work, too. A revised paper will have to admit this. R criticize also long per-scene train, but AC sees this only as a practical issue.

Rebuttal gave some answers to specific questions and some new experimental evidence. Could not sway ``yNkC``. One R was happy with the authors comments and increased their score. Some negative R's did not take part in the decision. One R was so kind and honest to admit that they only understand the method after the author response and changed to 'accept'.

The AC suggests the paper admits the long training time as a limitation and that it is more about the idea than about state-of-the-art performance.

Overall, vibes are positive and on a positive trajectory after the author response: Accept.